# Human perivascular stem cell-derived extracellular vesicles mediate bone repair

Jiajia Xu[1], Yiyun Wang[1], Ching-Yun Hsu[1], Yongxing Gao[1], Carolyn Ann Meyers[1], Leslie Chang[1], Leititia Zhang[1,2], Kristen Broderick[3], Catherine Ding[4], Bruno Peault[4,5,6], Kenneth Witwer[7,8], Aaron Watkins James[1]*

[1]Department of Pathology, Johns Hopkins University, Baltimore, United States; [2]Department of Oral and Maxillofacial Surgery, School of Stomatology, China Medical University, Shenyang, China; [3]Department of Surgery, Johns Hopkins University, Baltimore, United States; [4]Department of Orthopaedic Surgery, Orthopaedic Hospital Research Center, UCLA, Orthopaedic Hospital, Los Angeles, United States; [5]Centre For Cardiovascular Science, University of Edinburgh, Edinburgh, United Kingdom; [6]MRC Centre for Regenerative Medicine, University of Edinburgh, Edinburgh, United Kingdom; [7]Department of Molecular and Comparative Pathobiology, Johns Hopkins University, Baltimore, United States; [8]Department of Neurology, Johns Hopkins University, Baltimore, United States

**Abstract** The vascular wall is a source of progenitor cells that are able to induce skeletal repair, primarily by paracrine mechanisms. Here, the paracrine role of extracellular vesicles (EVs) in bone healing was investigated. First, purified human perivascular stem cells (PSCs) were observed to induce mitogenic, pro-migratory, and pro-osteogenic effects on osteoprogenitor cells while in non-contact co-culture via elaboration of EVs. PSC-derived EVs shared mitogenic, pro-migratory, and pro-osteogenic properties of their parent cell. PSC-EV effects were dependent on surface-associated tetraspanins, as demonstrated by EV trypsinization, or neutralizing antibodies for CD9 or CD81. Moreover, shRNA knockdown in recipient cells demonstrated requirement for the CD9/CD81 binding partners IGSF8 and PTGFRN for EV bioactivity. Finally, PSC-EVs stimulated bone repair, and did so via stimulation of skeletal cell proliferation, migration, and osteodifferentiation. In sum, PSC-EVs mediate the same tissue repair effects of perivascular stem cells, and represent an 'off-the-shelf' alternative for bone tissue regeneration.
DOI: https://doi.org/10.7554/eLife.48191.001

*For correspondence:
awjames@jhmi.edu

## Introduction

Stromal progenitor cells within vessel walls have multipotent properties (*Cathery et al., 2018*; *Corselli et al., 2012*; *Covas et al., 2008*; *Crisan et al., 2008*; *Dellavalle et al., 2007*; *Farrington-Rock et al., 2004*), are native forerunners of mesenchymal stem cells (MSCs), and participate in endogenous bone repair (*Diaz-Flores et al., 1991*; *Diaz-Flores et al., 1992*; *Grcevic et al., 2012*). When purified based on expression of CD146 (Mel-CAM) and CD34, human perivascular stem cells (PSCs) from adipose tissue or other tissue compartments speed bone repair (*Askarinam et al., 2013*; *Chung et al., 2014*; *James et al., 2017a*; *James et al., 2012a*; *James et al., 2012b*; *Tawonsawatruk et al., 2016*). Although direct incorporation of human PSCs into chondroblasts, osteoblasts and osteocytes occurs (*James et al., 2012b*), PSCs induce bone healing either primarily or exclusively via paracrine stimulation of the resident host cells within the defect niche (*Chung et al., 2014*; *Tawonsawatruk et al., 2016*). High expression and secretion of osteoinductive proteins has been observed among freshly sorted or culture-expanded PSCs, including bone

**eLife digest** Throughout our lives, our bodies need to heal after injury. Blood vessels are found throughout the body's tissues and are a source of cells that guide the process of repair. These cells, called perivascular stem cells (PSCs), are a type of stem cell found in the lining of blood vessels. Stem cells are cells that can become one of several different types of mature cells, depending on what the body needs.

Extracellular vesicles are bundles of chemical signals that cells send into their external environment. Just like an address or a tag on a parcel, specific molecules mark the exterior surface of these bundles to deliver the message to the right recipient. Stem cells often use extracellular vesicles to communicate with surrounding cells.

One role of PSCs is repairing damage to bones. Unusually, they do not turn into new bone cells and so do not directly contribute to the re-growing tissue. Instead, PSCs act indirectly, by stimulating the cells around them. How PSCs send these 'repair instructions' has, however, remained unclear. Xu et al. wanted to determine if PSCs used extracellular vesicles to direct bone repair, and if so, what 'tags' needed to be on the vesicles and on the receiving cells for this to happen.

Experiments using PSCs and immature bone cells grown in the laboratory allowed the PSCs' effect on bone cells to be simulated in a Petri dish. The two types of cells were grown on either side of a barrier, which separated them physically but allowed chemical signals through. In response to the PSCs, the immature bone cells multiplied, started to move (which is something they need to do to heal damaged tissue), and began to resemble mature bone cells.

Analysis of the signals released by the PSCs revealed that these were indeed extracellular vesicles, and that they were tagged by specific proteins called tetraspanins. Genetic manipulation of the immature bone cells later showed that these cells needed specific 'receiver' molecules to respond to the PSCs. Adding only extracellular vesicles to the bone cells, without any PSCs, confirmed that it was indeed the vesicles that triggered the healing response. Finally, giving the vesicles to mice with bone damage helped them to heal faster than untreated animals.

These results have uncovered a key mechanism by which stem cells control the repair of bone tissue. This could one day lead to better treatments for patients recovering from fractures or needing bone surgery.

DOI: https://doi.org/10.7554/eLife.48191.002

morphogenetic proteins (BMPs) and vascular endothelial growth factor (VEGF) among others (*Chen et al., 2009*; *Hardy et al., 2017*; *James et al., 2017b*). However, the exact paracrine intermediaries of PSC-induced bone defect healing are not known.

Extracellular vesicles (EVs), including exosomes and microvesicles, carry a repertoire of bioactive molecules: proteins, nucleic acids, lipids and carbohydrates (*Colombo et al., 2014*; *Cvjetkovic et al., 2016*; *Mateescu et al., 2017*). The heterogenous nature of EVs is well described, with EVs of a diameter <100–150 nm and multivesicular body (MVB) origin termed exosomes (*Gould and Raposo, 2013*; *Mateescu et al., 2017*). Stromal progenitor cells elaborate EVs, and EVs have been observed to harbor many of the regenerative properties of their parent cell (*Lakhter and Sims, 2015*). In bone repair, CD9 knockout mice, which have impaired EV elaboration, have delayed appendicular bone healing, that may be rescued by mesenchymal progenitor cell-derived EVs (*Furuta et al., 2016*). As well, induced pluripotent stem cell-derived EVs have been shown to incite osteogenesis and vasculogenesis in vivo (*Qin et al., 2016*; *Todorova et al., 2017*). The potential role of miRNA cargo in EVs has been implicated in these effects, however the mechanisms that underlie EV-mediated reparative effects in bone are essentially unknown.

Here, we observe that in similarity to their parent cell, PSC-derived EVs incite pleiotropic effects on skeletal progenitor cells, including mitogenic, pro-migratory, pro-osteogenic effects, and broad ranging changes in the cellular transcriptome of the recipient cell. These bioactive effects require EV tetraspanin activity, as well as expression of their binding partners IGSF8 and PTGFRN by the recipient progenitor cell. These cellular effects of PSC-EVs on skeletal progenitor cells converge to incite intramembranous bone repair in a mouse model.

## Results

### Perivascular stem cells induce paracrine effects on BMSCs via EV release

Perivascular stem/stromal cells (PSCs) were derived from human white adipose tissue using previously reported methods, by FACS selection of the related populations of CD34$^+$CD146$^-$CD45$^-$CD31$^-$ adventitial progenitor cells and CD146$^+$CD34$^-$CD45$^-$CD31$^-$ pericytes (*Corselli et al., 2012*; *Crisan et al., 2008*; *James et al., 2017a*; *Meyers et al., 2018aMeyers et al., 2018b*; *Meyers et al., 2018b*) (*Figure 1—figure supplement 1*). Frequencies of human PSCs within adipose tissue are summarized in *Supplementary file 2*, and were within our previously reported ranges (*James et al., 2012a*; *West et al., 2016*). In order to better understand the paracrine effects of PSCs in bone repair, we first defined a set of replicable in vitro paracrine effects of human PSCs on recipient human BMSCs (*Figure 1*). Prior to all experiments, cell surface antigens of human PSCs and BMSCs were typified by flow cytometry, and the multilineage differentiation potential of BMSCs was confirmed (*Figure 1—figure supplement 2* and *Figure 1—figure supplement 3*).

Co-culture results showed that PSCs induced significant mitogenic (*Figure 1A*, MTS assay), pro-migratory (*Figure 1B*, scratch wound healing assay), and pro-osteogenic effects (*Figure 1C*, alkaline phosphatase activity) on BMSCs when placed in non-contact co-culture conditions. When a lipophilic dye (PKH26) was used to label the cell membrane of PSCs, this dye was incorporated into the BMSCs, suggesting the exchange of membrane from parent PSCs to recipient cells under non-contact culture conditions (*Figure 1D*). As primary candidates for mediating non-contact dependent cell-cell communication, PSC-derived extracellular vesicles (EVs) were examined (*Figure 1E–H*). PSC-EVs were enriched by ultracentrifugation of serum-free supernatant of PSCs. Guidelines from the International Society of Extracellular Vesicles (ISEV) for EV characterization were observed (*Lötvall et al., 2014*). Spherical PSC-EV morphology was observed by transmission EM (*Figure 1E*), with enrichment of tetraspanin molecule expression (CD9, CD63, and CD81), but without expression of the endoplasmic reticulum protein calnexin (*Figure 1F*). Size distribution of PSC-EVs was most commonly ~100 nm, as shown by either TEM quantification (*Figure 1G*) or nanoparticle tracking analysis (NanoSight) (*Figure 1H*). EV yield per cell per day as assessed by protein content was 1.11 ± 0.18 pg, as determined by the Bradford method (*Supplementary file 3*). Thus, PSCs elaborate EVs in culture which then are received by stromal progenitor cells, representing a primary candidate for PSC-mediated paracrine activity.

EVs represent a mixture of protein, lipid, and RNA that may affect cellular processes of the recipient cell (*Bidarimath et al., 2017*). To begin to examine this, total RNA sequencing was performed on three PSC-EV preparations and compared to the RNA content of their parent PSC cells (*Figure 1I–K*). To determine which genes were most expressed, transcripts were normalized by fragments per kilobasepair per million mapped (FPKM), and those with Log2 FPKM >−0.8 underwent further analysis. Among these, 10,256 annotated genes were expressed in all samples of 54,136 total RNA transcripts (19% of total, including 10,256 protein coding RNA; six non-coding RNA; and four pseudo RNA). Clear separation between gene expression profiles were observed when comparing PSC-EV RNA content to their parent cells, as observed by unsupervised hierarchical clustering (*Figure 1I*) and principal component analysis (*Figure 1J*). Putative gene markers of human perivascular MSC were cross-referenced within PSCs and PSC-EVs (*Cho et al., 2017*). Higher expression of most transcripts were seen in each PSC isolate (such as *CD44, NT5E, ENG, LEPR, PDGFRA*, and *PDGFRB*), with comparatively less expression among PSC-EVs (*Figure 1K*). The highest expressing 100 genes within PSC-EVs are listed in *Supplementary file 4*. All 100 were protein coding genes, 57 encoded for ribosomal proteins (e.g. *RPS18, RPL37A*, and *RPL41*) and 68 were included within Gene Ontology term extracellular exosome (GO term: 0070062). Reflecting their adipose tissue origin, these highly expressed PSC-EV transcripts included several genes associated with adipocytes or adipogenesis, such as *VIM, RACK1*, and *LGALS1*. Highly expressed PSC-EV genes also included transcripts involved in the regulation of cellular proliferation (e.g. *FTL, RAB13, MT2A*, and *TMSB10*) or cellular migration (e.g. *RAB13, TMSB10, S100A6*, and *EEF1A1*). Next, PSC-EV RNA content was cross-referenced to a previously published RNA Seq dataset within unpurified adipose stromal cell-derived EVs from porcine tissue (*Eirin et al., 2014*). As expected, some overlap existed in EV content between these two adipose stromal cell derivatives (*Supplementary file 5*). For example, of 39 transcription factors previously found to be enriched in adipose tissue stem/stromal cell (ASC)-EVs

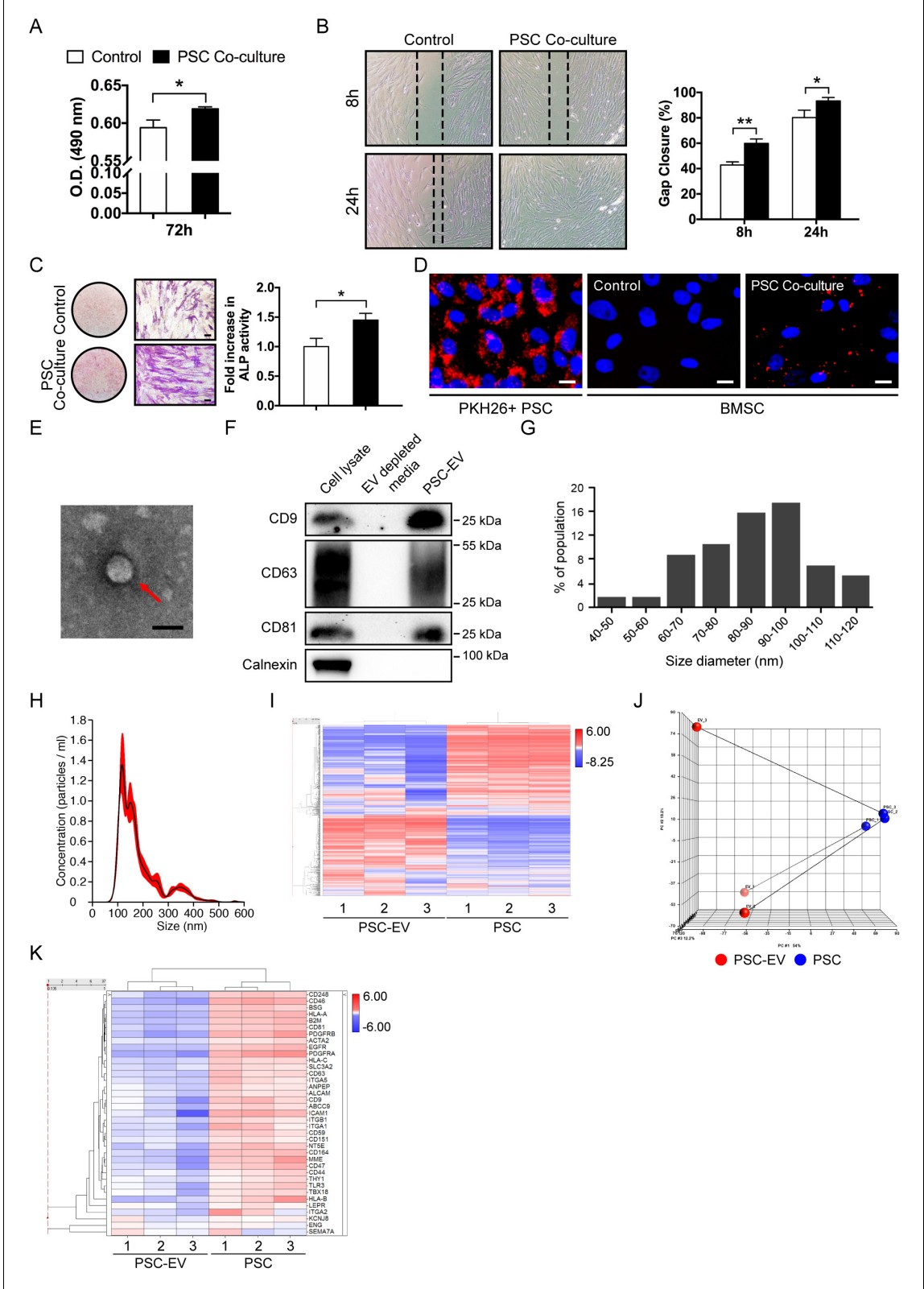

**Figure 1.** Perivascular stem cells (PSCs) promote BMSC proliferation, migration, and osteogenesis with extracellular vesicle (EV) elaboration in non-contact co-culture. Human adipose tissue-derived perivascular stem cells (PSCs) were placed in transwell inserts and the effects on human BMSCs were assessed. (A) MTS assay among BMSCs after 72 hr with or without PSCs co-culture. (B) Migration assay among BMSCs after 8 and 24 hr with or without PSCs co-culture. Representative 100x images above, with percentage gap closure below. (C) Alkaline phosphatase staining (left, whole well images and

*Figure 1 continued on next page*

*Figure 1 continued*

a high magnification view) and quantification (right) among BMSCs after 72 hr with or without PSCs co-culture. (D) BMSCs were cultured with or without PKH26 (red)-labeled PSCs in co-culture. Images (200x) after 48 hr, with DAPI nuclear counterstain (blue). (E) Representative transmission electron microscopy image of PSC-derived extracellular vesicles (PSC-EVs). (F) Western blot for tetraspanin expression (CD9, CD63, and CD81) among PSC cell lysate, EV-depleted supernatant, and purified PSC-EVs. In comparison, the endoplasmic reticulum-associated protein calnexin was also assayed. (G,H) Size distribution of PSC-EVs based on (G) electron microscopy image analysis and (H) Nanoparticle tracking analysis (NanoSight). (I–K) Total RNA sequencing comparison of PSC-EVs to their parent PSCs. (I) Unsupervised hierarchical clustering among PSC-EVs and PSCs. (J) Principal component analysis among PSC-EVs and PSCs. Lines match PSC-EVs to their respective parent cells. (K) Heat map demonstrating mRNA expression levels of putative perivascular markers among PSC-EVs and PSCs. PSC: perivascular stem cell; PSC-EV: perivascular stem cell-derived extracellular vesicle; BMSC, bone marrow mesenchymal stem/stromal cell. Data shown as mean ± SD, and represent triplicate experimental replicates in biological duplicate. White scale bar: 20 μm. Black scale bar: 100 nm. *p<0.05; **p<0.01.

DOI: https://doi.org/10.7554/eLife.48191.003

The following figure supplements are available for figure 1:

**Figure supplement 1.** Human PSC derivation by fluorescence activated cell sorting.

DOI: https://doi.org/10.7554/eLife.48191.004

**Figure supplement 2.** Flow cytometry analysis of FACS derived human PSC.

DOI: https://doi.org/10.7554/eLife.48191.005

**Figure supplement 3.** Flow cytometry analysis and multilineage differentiation potential of human culture-derived BMSC.

DOI: https://doi.org/10.7554/eLife.48191.006

(*Eirin et al., 2014*), 14 genes (35.9%) were also enriched in human PSC-EVs. Of these, known positive regulators of cellular proliferation (e.g. *JMJD1C*, *NRIP1*, and *TRPS1*) and cellular migration (e.g. *JMJD1C*, *TCF4*, and *KLF7*) were identified. Of note, several transcriptional repressors were found within the previously published ASC-EVs (e.g. *ZNF568*, *ZHX1*, *ZBTB1*, and *RUNX1T1*), which were not found at high levels in human PSC-EVs. Thus, PSC-EVs demonstrate both commonalities and distinct differences in RNA content when compared to their either parent cells, as well as to known expression profiles among unpurified adipose stromal EVs.

## PSC-EVs promote BMSC proliferation, migration, and osteogenic differentiation

The direct effects of PSC-EVs on recipient BMSCs were next defined (*Figure 2*). To demonstrate interaction of EVs with the recipient cells, PSC-EVs were first labeled with PKH26 dye, and fluorescently labeled PSC-EVs were directly applied to human BMSCs. After 48 hr, fluorescence was observed on the recipient BMSCs in a cytosolic distribution (*Figure 2A*). PSC-EVs were next labeled with a pH dependent dye (pHrodo Red Maleimide), which only fluoresces after exposure to an acidic environment such as the cellular interior (*Figure 2B*). After 48 hr, fluorescence was again observed suggesting EV internalization by the recipient BMSCs. PSC-EVs exerted a dose-dependent mitogenic effect on recipient BMSCs (*Figures 2C* and *1*, 2.5, and 5 μg/mL, MTS assays). Using a scratch wound healing assay, PSC-EVs represented a significant pro-migratory stimulus, with the lowest PSC-EV concentration demonstrating the largest effect (*Figure 2D*). Under osteogenic differentiation conditions, PSC-EVs induced a dose dependent increase in ALP activity (*Figure 2E*) and bone nodule deposition (*Figure 2F*). Quantitative PCR analysis of characteristic osteoblastic gene markers confirmed a dose-dependent positive regulatory effect by PSC-EVs (*Figure 2G*), including *RUNX2 (Runt related transcription factor 2)* and *SP7 (Osterix)*. We next queried as to whether recipient stromal cells responded to PSC-EVs in a tissue-specific manner. Here, human culture-derived ASCs were exposed to the identical PSC-EV treatment conditions with broad similarities and minor differences found (*Figure 2—figure supplement 1*). Unlike BMSCs, the mitogenic effect of PSC-EVs was not observed among recipient ASCs. The pro-migratory and pro-osteogenic effects of PSC-EVs were conserved findings across both recipient stromal cells, although the maximum efficacious dose for each bioactivity assay was different when comparing BM- and AT-derived recipient cells. Finally, we compared the effects of EVs from unsorted ASCs and homogeneous PSCs on recipient BMSCs. Like PSC-EVs, ASC-EVs exerted mitogenic, pro-migratory, and pro-osteogenic effects on recipient BMSCs (*Figure 2—figure supplement 2*). However, the magnitude of change was significantly different between EV preparations, and PSC-EVs demonstrated a stronger induction of cell proliferation and osteogenic differentiation, while ASC-EVs enhanced cell migration to a greater degree. In

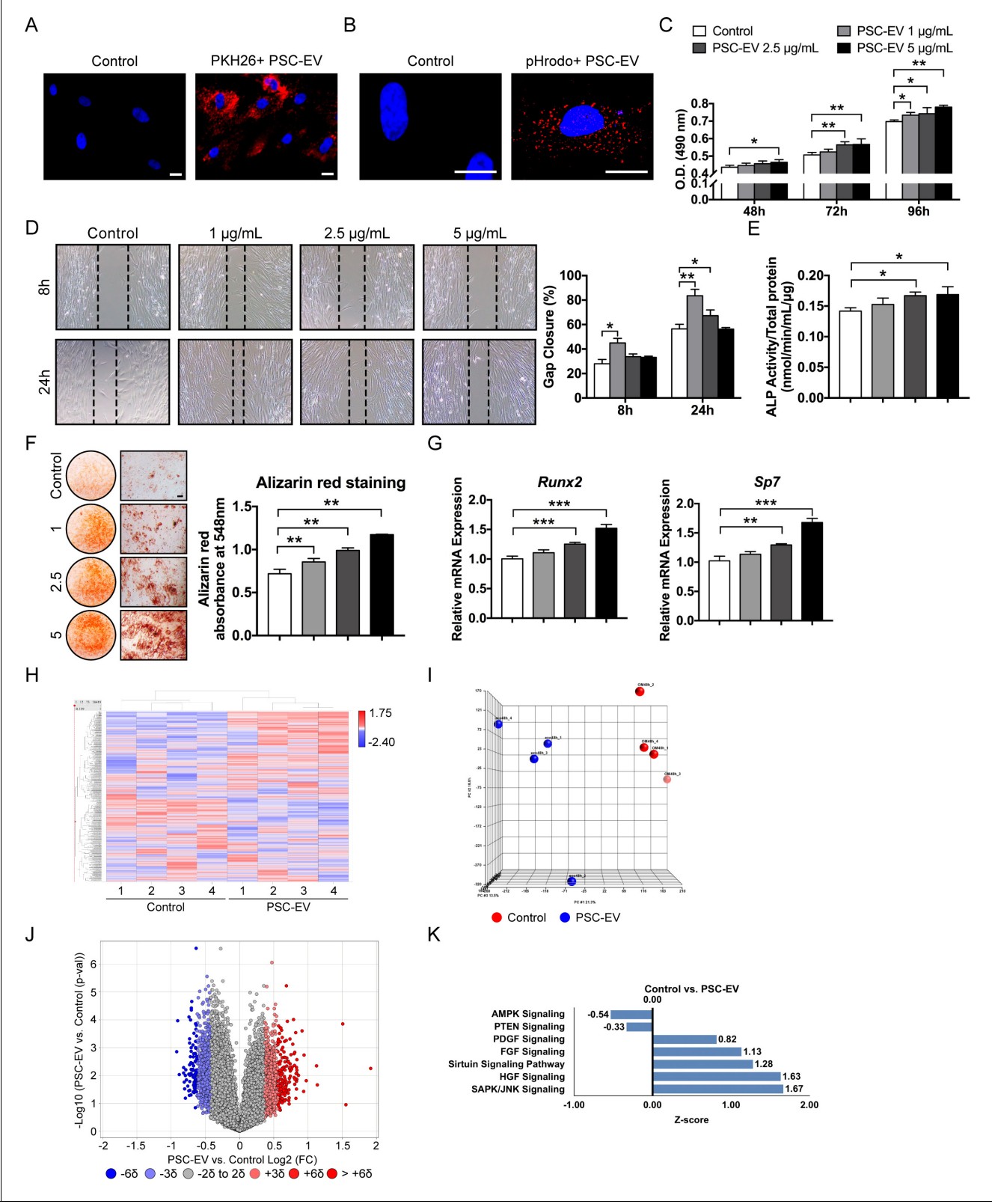

**Figure 2.** PSC-EVs promote BMSC proliferation, migration, and mineralization. (**A**) Appearance of BMSCs treated with or without PKH26 (red)-labeled PSC-EVs. (**B**) Appearance of BMSCs treated with or without pHrodo (red)-labeled PSC-EVs. Images after 48 hr, with DAPI nuclear counterstain. (**C**) BMSC proliferation assessed by MTS assay at 48, 72, and 96 hr with or without PSC-EVs (1–5 µg/mL). (**D**) BMSC migration assessed by scratch wound healing assay at 8 and 24 hr with or without PSC-EVs (1–5 µg/mL). Representative 100x images with percentage gap closure are shown. (**E**) Alkaline

*Figure 2 continued on next page*

*Figure 2 continued*

phosphatase (ALP) activity of BMSCs at 3 days in osteogenic medium (OM) treated with or without PSC-EVs (1–5 µg/mL). (F) BMSC osteogenic differentiation assessed by Alizarin Red staining at 7 days in osteogenic medium (OM) with or without PSC-EVs (1–5 µg/mL). Photometric quantification shown right. (G) BMSC osteogenic gene expression by qPCR at 7 days in OM with or without PSC-EVs (1–5 µg/mL), including *Runt related transcription factor 2* (*Runx2*) and *Sp7* (*Osterix)*. (H–K) PSC-EVs induced changes in the BMSC transcriptome, as assessed by Clariom D microarray. BMSCs were cultured in the presence or absence of PSC-EVs (2.5 µg/mL) for 48 hr. (H) Unsupervised hierarchical clustering among BMSCs treated with or without PSC-EVs. (I) Principal component analysis among BMSCs treated with or without PSC-EVs. (J) Volcano plot of all transcripts. X-axis represents Log2 fold change for each gene. Y-axis represents -Log10 p value. Red dots indicate >2 SD increase among PSC-EV-treated samples. Blue dots indicate >2 SD decrease among PSC-EV-treated samples. (K) Ingenuity Pathway Analysis (IPA) identified representative pathways that were upregulated (Z-score >0) or downregulated (Z-score <0) among BMSCs treated with PSC-EVs. PSC: perivascular stem cell; PSC-EV: perivascular stem cell-derived extracellular vesicle; BMSC, bone marrow mesenchymal stem cell. Data shown as mean ± SD, and represent triplicate experimental replicates. White scale bar: 20 µm. Black scale bar: 100 nm. *p<0.05; **p<0.01; ***p<0.001.

DOI: https://doi.org/10.7554/eLife.48191.007

The following figure supplements are available for figure 2:

**Figure supplement 1.** PSC-EV promote ASC migration and mineralization.

DOI: https://doi.org/10.7554/eLife.48191.008

**Figure supplement 2.** Comparative effects of human ASC-EVs and PSC-EVs on BMSC proliferation, migration, and osteogenesis.

DOI: https://doi.org/10.7554/eLife.48191.009

**Figure supplement 3.** Distribution of PSC-EV associated transcripts within control- or EV-treated BMSC.

DOI: https://doi.org/10.7554/eLife.48191.010

sum, perivascular EVs retain bioactive effects of their parent perivascular cell type and exert overall similar effects across different recipient multipotent mesenchymal cell types.

Next, the changes in the transcriptome of the recipient BMSCs were examined. Here, the Affymetrix Clariom D microarray assayed the BMSC transcriptome at 48 hr post-PSC-EV treatment (*Figure 2H–K*). 38,416 annotated genes were expressed in all samples among 135,750 total probesets (20,014 protein coding RNA; 5054 non-coding RNA; 2795 pseudo RNA; 207 sno RNA; 55 snRNA; 26 rRNA; two scRNA). Clear differences between control- and PSC-EV-treated BMSC gene expression were observed by unsupervised hierarchical clustering (*Figure 2H*) and principal component analysis (*Figure 2I*). As shown by volcano plot (*Figure 2J*), 4129 transcripts showed >2 SD increase in expression with PSC-EV treatment (3.04% of total, red dots), while 2589 transcripts showed >2 SD reduction in expression with PSC-EV treatment (1.91% of total, blue dots). QIAGEN Ingenuity Pathway Analysis (IPA) showed that the majority of the activated pathways were associated with the positive regulation of cell proliferation, migration, and/or osteogenesis, including for example SAPK/JNK signaling (*Zha et al., 2016*), HGF signaling (*Forte et al., 2006*), Sirtuin signaling (*Simic et al., 2013*), FGF Signaling (*Gharibi and Hughes, 2012*), and PDGF Signaling (*Li et al., 2014*), among recipient BMSCs (*Figure 2K*, see *Supplementary file 6* for a complete list). Conversely, downregulated signaling pathways among recipient BMSCs included those known to negatively regulate cell proliferation, such as AMPK Signaling (*de Meester et al., 2014*). PTEN signaling was also downregulated, which negatively regulates BMSC proliferation (*Shen et al., 2018*), migration (*Comer and Parent, 2002*) and osteogenic differentiation (*Liu et al., 2017*) (*Figure 2K*, see *Supplementary file 7* for a complete list). Next, the transcriptome of PSC-EVs (as previously determined by RNA-Seq) was cross-referenced to changes within the BMSC transcriptome after EV treatment (*Figure 2—figure supplement 3*). Here, of 7789 PSC-EV transcripts with a mean FPKM >0, approximately half of these transcripts demonstrated were increased among PSC-EV-treated BMSCs (3678 transcripts, 47.22% of total). Thus, global changes in the recipient BMSC transcriptome were not well explained by simple transfer of EV RNA cargo. Instead, PSC-EVs demonstrate prominent regulation of recipient cell gene transcription resulting in significant mitogenic, pro-migratory, and pro-osteogenic effects in vitro.

## EV surface-associated tetraspanins interact with recipient cell binding partners for bioactivity

EV uptake and downstream bioactive effects may be dependent on membrane-bound protein interaction with the recipient cell (*Escrevente et al., 2011*; *Janas et al., 2016*; *Morelli et al., 2004*). To assess this, trypsinization was performed to digest EV membrane-bound proteins and the cellular

effects of trypsinized PSC-EVs were again assessed (*Figure 3—figure supplements 1* and 2.5 μg/mL PSC-EVs used). Pretreatment with trypsin completely abrogated the mitogenic (*Figure 3—figure supplement 1A*), pro-migratory (*Figure 3—figure supplement 1B*), and pro-osteogenic effects of PSC-EVs (*Figure 3—figure supplement 1C*). Thus, surface-associated vesicular proteins are integral for PSC-EV bioactivity.

Tetraspanins are enriched among exosomes including PSC-EVs (see again *Figure 1F*), and both CD9 and CD81 are fusogenic, involved in spermatozoa and phagocyte fusion (*Rubinstein et al., 2006a*; *Rubinstein et al., 2006b*; *Takeda et al., 2003*; *van Dongen et al., 2016*; *Zhu et al., 2002*). In contexts outside mesenchymal progenitor cell biology, antibodies against tetraspanins have mitigated EV interaction with the recipient cell (*Morelli et al., 2004*). To test the requirement of surface-associated tetraspanins, neutralizing antibodies directed against CD9 or CD81 were pre-incubated with PSC-EVs before application to the recipient cell (*Figure 3*). PSC-EVs were labeled with a pH dependent dye and treated with or without neutralizing antibodies (*Figure 3*). After 48 hr and in

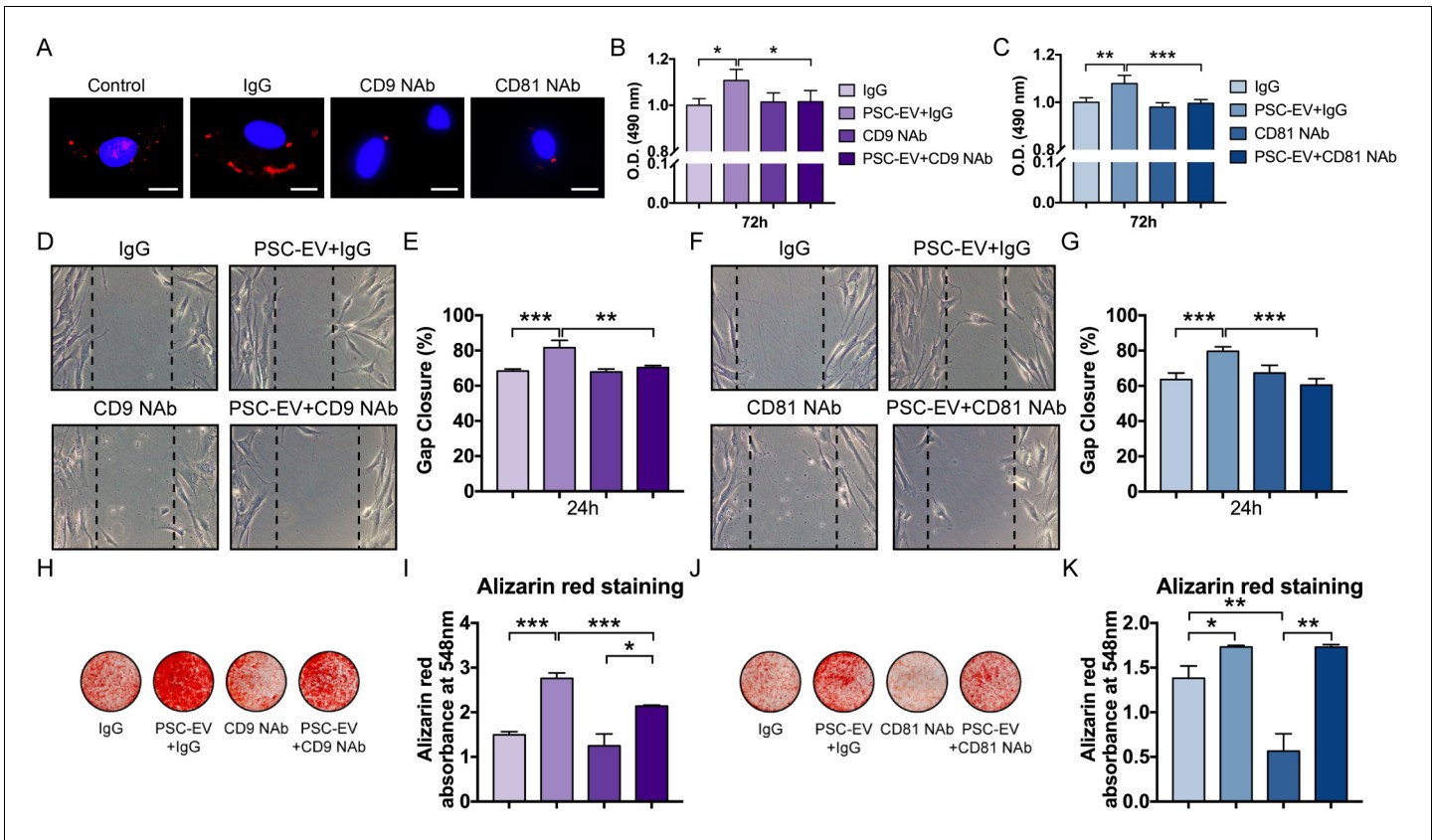

**Figure 3.** PSC-EVs require tetraspanins for bioactive effects on BMSCs. (A) Appearance of BMSCs treated with pHrodo (red)-labeled PSC-EVs in the context of incubation with neutralizing antibodies to CD9, CD81, or isotype control (IgG). Images after 48 hr, with DAPI nuclear counterstain. White scale bar: 15 μm. (B,C) BMSC proliferation assessed by MTS assay at 72 hr, with or without anti-CD9 (B) or anti-CD81 (C) neutralizing antibodies. (D,E) BMSC migration assessed by scratch wound healing assay at 8 hr with or without anti-CD9 neutralizing antibodies, shown by microscopic images (D) and quantification (E). (F,G) BMSC migration assessed by scratch wound healing assay at 8 hr with or without anti-CD81 neutralizing antibodies, shown by microscopic images (F) and quantification (G). (H,I) BMSC osteogenic differentiation with or without anti-CD9 neutralizing antibodies, as assessed by Alizarin Red staining (H) and photometric quantification (I) at 7 days in OM. (J,K) BMSC osteogenic differentiation with or without anti-CD81 neutralizing antibodies, as assessed by Alizarin Red staining (J) and photometric quantification (K) at 7 days in OM. PSC: perivascular stem cell; PSC-EV: perivascular stem cell-derived extracellular vesicle; BMSC, bone marrow mesenchymal stem cell. Data shown as mean ± SD, and represent triplicate experimental replicates. *p<0.05; **p<0.01; ***p<0.001.

DOI: https://doi.org/10.7554/eLife.48191.011

The following figure supplement is available for figure 3:

**Figure supplement 1.** PSC-EVs require surface-associated proteins for bioactive effects on BMSCs.
DOI: https://doi.org/10.7554/eLife.48191.012

contrast to control conditions, minimal fluorescence was seen with either neutralizing antibody, suggesting that CD9 and CD81 were required for EV internalization (*Figure 3A*). Results demonstrated that both anti-CD9 and anti-CD81 reversed the mitogenic effects of PSC-EVs (*Figure 3B,C*). Likewise, the pro-migratory effects of PSC-EVs were completely abrogated by either neutralizing antibody (*Figure 3D–G*). The pro-osteogenic effects of PSC-EVs were partially reversed by anti-CD9 (*Figure 3H,I*), but not anti-CD81 neutralizing antibodies (*Figure 3J,K*). To explore other EV-associated proteins that may be important for PSC-EV bioactivity, we analyzed other CD markers that were enriched in PSC-EVs (mean FPKM >0; see *Supplementary file 8*). Several additional fusogenic proteins were identified, including *CD46* and *CD63* (*Anderson et al., 2004*; *Raaben et al., 2017*). In addition, several other CD markers were enriched among PSC-EV that have described pro-osteogenic effects, including *CD44*, *CD82*, and *CD99* (*Bergsma et al., 2018*; *Oranger et al., 2015*; *Yeh et al., 2014*). In sum, intact activity of CD9 or CD81 are essential for the majority of bioactive effects of PSC-EVs on recipient osteoprogenitor cells.

CD9 and CD81 are known to interact with several cell surface proteins on the recipient cell, including immunoglobulin superfamily, member 8 (IGSF8) (*Glazar and Evans, 2009*) and prostaglandin F2 receptor inhibitor (PTGFRN) (*Charrin et al., 2001*). In a candidate fashion, shRNA-mediated knockdown of *IGSF8* and *PTGFRN* was performed in human recipient BMSCs (*Figure 4A*, 52.9% and 58.4% knockdown of *IGSF8* and *PTGFRN* gene expression, respectively). PSC-EVs were labeled with a pH-dependent dye and incubated with BMSCs with or without knockdown. After 48 hr and unlike vector control conditions, no fluorescence was seen among BMSCs with *IGSF8* or *PTGFRN* knockdown (*Figure 4B*). In comparison to vector control, knockdown of either *IGSF8* or *PTGFRN* abrogated the mitogenic effects of PSC-EVs (*Figure 4C*, 2.5 µg/mL PSC-EVs used). Likewise, *IGSF8* or *PTGFRN* KD nullified the pro-migratory effect of PSC-EVs (*Figure 4D*). Finally, *PTGFRN* KD inhibited the pro-osteogenic effect of PSC-EVs (*Figure 4E*). In contrast, *IGSF8* KD paradoxically increased BMSC osteogenic differentiation; however, this was observed in both control and PSC-EV treatment

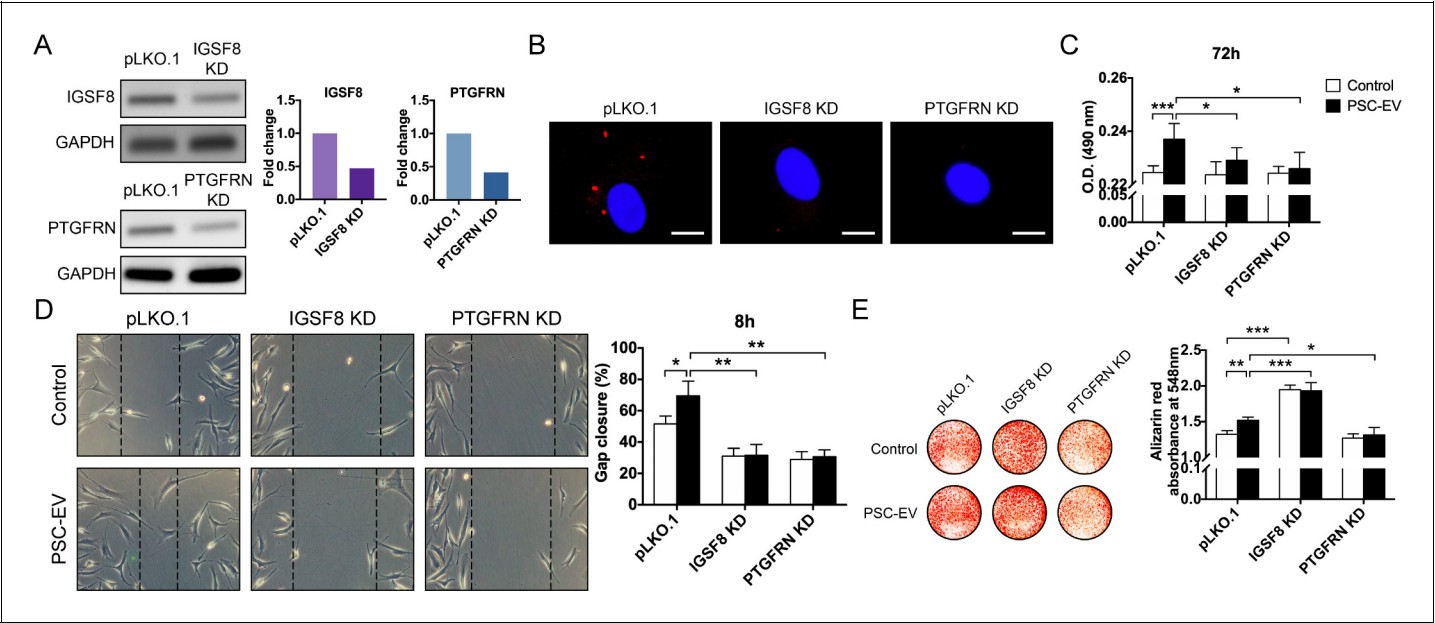

**Figure 4.** PSC-EV bioactivity requires IGSF8 and PTGFRN expression on recipient BMSCs. (**A**) Gene expression of either *IGSF8* or *PTGFRN* as assessed by regular PCR after shRNA mediated knockdown (96 hr shown). (**B**) Appearance of *IGSF8* or *PTGFRN* shRNA silenced BMSCs treated with pHrodo (red)-labeled PSC-EVs. Images after 48 hr, with DAPI nuclear counterstain. White scale bar: 15 µm. (**C**) BMSC proliferation assessed by MTS assay at 72 hr with or without *IGSF8* or *PTGFRN* shRNA, with or without PSC-EV treatment (2.5 µg/mL). (**D**) BMSC migration assessed by scratch wound healing assay at 8 hr with or without *IGSF8* or *PTGFRN* shRNA, with or without PSC-EV treatment (2.5 µg/mL). (**E**) BMSC osteogenic differentiation assessed by Alizarin Red staining and photometric quantification at 7 days in OM. Data shown as mean ± SD, and represent triplicate experimental replicates. PSC: perivascular stem cell; PSC-EV: perivascular stem cell-derived extracellular vesicle; BMSC, bone marrow mesenchymal stem cell. *p<0.05; **p<0.01; ***p<0.001.
DOI: https://doi.org/10.7554/eLife.48191.013

conditions and therefore appeared to be an EV independent phenomenon. Thus and in aggregate, EV-associated CD9 and CD81 along with their known binding partners on the recipient cell are required for the majority of bioactive effects of perivascular EVs.

## PSC-EVs induce proliferation, migration, and osteogenic differentiation to induce calvarial defect healing

Next, we investigated whether PSC-EV treatment would improve calvarial defect repair (*Figure 5*). The effects of the implanted parent PSC themselves on calvarial bone repair have been previously documented (*James et al., 2012a*). In order to confirm that human PSC-EVs would demonstrate bio-activity to mouse recipient osteoprogenitor cells, additional in vitro studies were first performed. Either mouse ASCs or neonatal mouse calvarial cells (NMCCs) were isolated and treated with or without PSC-EVs (1–5 µg/mL). In similarity to human recipient cells, PSC-EVs treatment resulted in similar mitogenic, pro-migratory, and pro-osteogenic effects on both mouse ASCs and NMCCs (*Figure 5—figure supplement 1*). To assess the ability of PSC-EVs to speed bone repair, we chose a bone injury model that shows some modest healing overtime (a 1.8 mm diameter, full thickness, circular frontal bone defect in the calvaria) (*Zhang et al., 2018*). Percutaneous injection of PSC-EVs (1 and 2.5 µg) was performed over the defect site twice weekly, and analyses were performed at 4 weeks postoperative. A summary of the animal treatment protocol is provided as *Figure 5—figure supplement 2*. A summary of animal numbers and treatment allocation is shown in *Supplementary file 9*. As in our in vitro studies, the effects of PSC-EVs on skeletal cell proliferation, migration, and osteodifferentiation were sequentially addressed. PSC-EV-treated defects showed an increase in osteoblastic proliferation at the bone defect edge, as shown by Ki67 immunostaining and quantitation of immunoreactive cells at the bone defect edge (*Figure 5A,B*, appearing red with arrowheads). In order to assess progenitor cell migration into the defect, a lineage tracing strategy was employed using Pdgfrα-CreER;eGFP animals (*Figure 5C*). Reporter activity within Pdgfrα-CreER;eGFP highlights bone-lining stromal cells which then migrate into populate the bone injury site. PSC-EV-treated defects showed a prominent increase in GFP[+] stromal progenitor cell migration into the defect mid-substance (*Figure 5C,D*, quantified right as the mean fluorescence intensity within the middle of the defect site). Microcomputed tomography (µCT) reconstructions revealed an increase in defect re-ossification in comparison to PBS control (*Figure 5E*). Quantitative indices confirmed an increase in bone healing across all metrics, including bone volume (BV, *Figure 5F*), bone formation area (BFA, *Figure 5G*), and semi-quantitative healing score (*Figure 5H*) (*Spicer et al., 2012*). Routine H&E staining confirmed a significant narrowing of the gap between bony fronts (*Figure 5I*, black arrowheads) and an enrichment of osteocalcin (OCN)+ cells at the leading edges of the defect site with PSC-EV treatment (*Figure 5J,K*). In sum, the mitogenic, pro-migratory, and osteoinductive effects of PSC-EVs converge in vivo to speed bone defect healing.

## Discussion

In sum, perivascular EVs induce proliferation, migration and osteogenic differentiation of skeletal progenitor cells, and the confluence of these effects positively regulates bone defect repair. Here, low doses of PSC-EVs represented a potent migratory stimulus, while higher doses of PSC-EVs induced proliferation and differentiation. Presumably this has in vivo relevance in a skeletal injury site where a gradient concentration of EVs may exist. Skeletal progenitor cells at a distance may be recruited with low concentrations of EVs, and under exposure to higher EV concentrations they may expand in numbers and deposit bone matrix.

Our study suggests several intriguing questions regarding the uniqueness of both the parent and recipient cell type in mesenchymal progenitor crosstalk and EVs. Other groups have shown that the parent cell type for EVs matters in bone biology. For example, 'MSC'-derived EVs assist in fracture healing, whereas osteosarcoma-derived EVs do not (*Furuta et al., 2016*). These findings may represent differential cargo between EVs. In terms of the parent cell in the current study, PSCs represent a FACS purified and relatively homogeneous stromal cell population in comparison to most current culture-based derivation protocols (*James et al., 2017a*). For this reason, PSCs may theoretically represent a more potent EV source as compared other stromal cell populations. Yet, ongoing and published studies from our research group suggest that PSCs exhibit profound intra-tissue cellular diversity. For example, *Hardy et al. (2017)* showed that ALDH (aldehyde dehydrogenase) bright

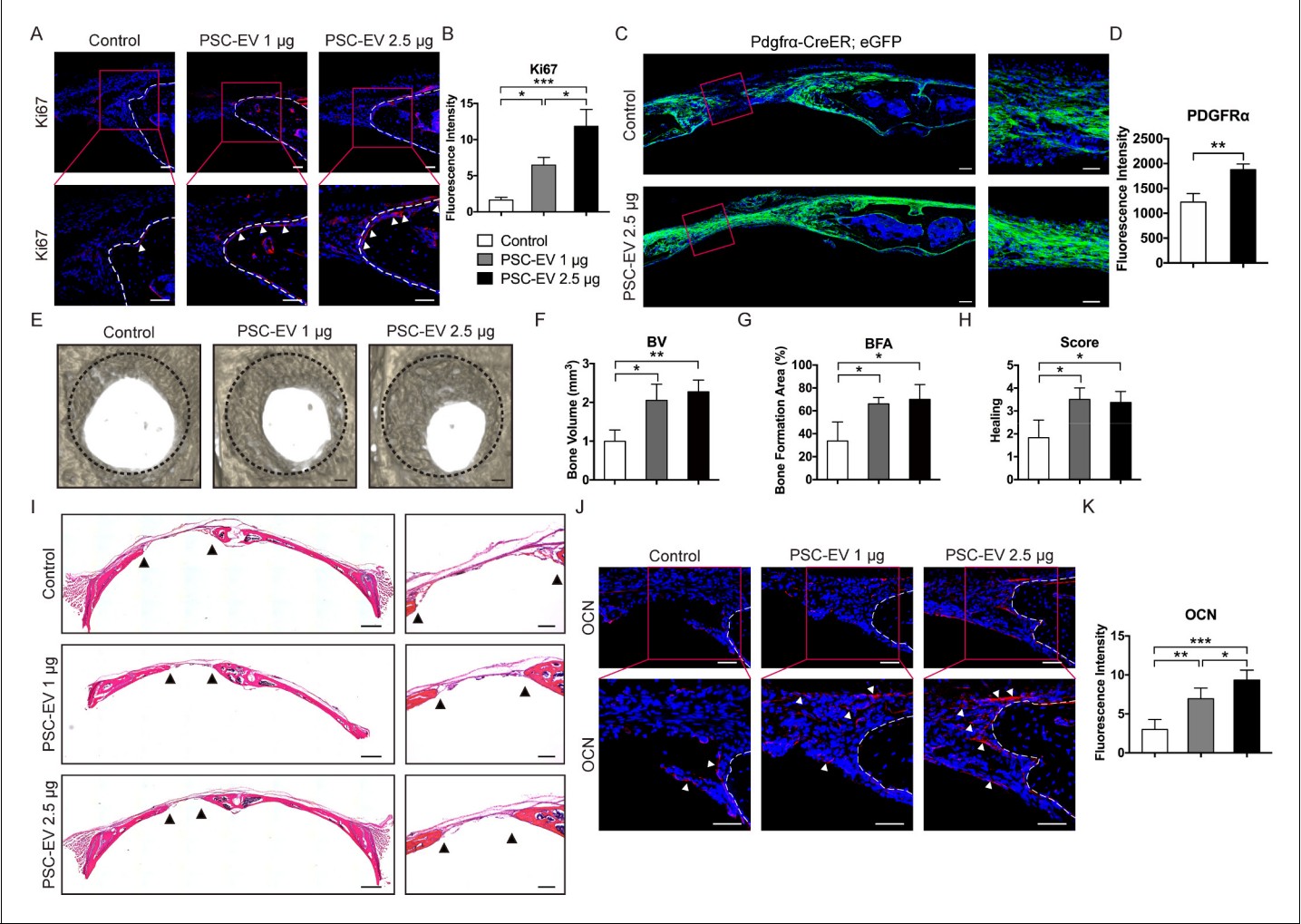

**Figure 5.** PSC-EVs promote calvarial bone regeneration in vivo. PSC-EVs (1 or 2.5 μg) were percutaneously delivered twice weekly overlying a circular, full thickness frontal bone defect site (1.8 mm diameter). Analysis was performed at 4 weeks thereafter. See also *Figure 5—figure supplement 2* for a schematic representation. (A) Cell proliferation at the bone defect edge (white arrowheads), as assessed by Ki67 immunofluorescent detection. White scale bar: 50 μm. (B) Quantification of bone-lining Ki67 immunoreactivity was shown. (C) Stromal/osteoprogenitor cell migration, as assessed by Pdgfrα-CreER; eGFP cell lineage tracing. Tile scan (left, scale bar: 100 μm) and high-magnification images of the defect site (right, scale bar: 50 μm) demonstrate migration of GFP[+] progenitor cells into the defect site. (D) Quantification of eGFP reporter activity in the mid-defect was shown. (E) Bone healing assessed by microcomputed tomography (μCT), shown in a top-down view. Black scale bar: 200 μm. (F–H) μCT analysis including (F) Bone Volume (BV), (G) bone formation area (BFA), and (H) healing score for the extent of bony bridging and bone union. (I) H&E appearance of the defect site. Tile scan (left, scale bar: 500 μm) and high-magnification images of the defect site (right, scale bar: 200 μm), with bone edges indicated by black arrowheads. (J) Osteocalcin (OCN) immunofluorescent detection at the bone defect edge. White scale bar: 50 μm. (K) Quantification of bone-lining OCN immunoreactivity was shown. PSC-EV: perivascular stem cell-derived extracellular vesicle; BMSC, bone marrow mesenchymal stem cell. Data shown as mean ± SD, and represent N = 4 defects per treatment group. *p<0.05; **p<0.01; ***p<0.001.
DOI: https://doi.org/10.7554/eLife.48191.014

The following figure supplements are available for figure 5:

**Figure supplement 1.** PSC-EVs promote mouse cell proliferation, migration, and mineralization.
DOI: https://doi.org/10.7554/eLife.48191.015
**Figure supplement 2.** Animal treatment protocol.
DOI: https://doi.org/10.7554/eLife.48191.016

and dim subpopulations of adipose tissue-derived perivascular cells were transcriptionally distinct, and this marker could be used to identify a more 'primitive' subpopulation. Likewise and in mice, functionally distinct pericyte subtypes have been examined using nestin and NG2 transgenic reporters (*Birbrair et al., 2014*; *Birbrair et al., 2013a*; *Birbrair et al., 2013b*). How perivascular progenitor

subpopulations exhibit overlapping or distinct vesicular secretomes is an interesting future avenue of investigation.

The recipient cell type is also important to consider in EV-based intercellular communication. In the present study, we focused explicitly on human BMSCs. Yet, our studies did examine adipose tissue progenitors and found basal differences in responsiveness to PSC-EVs. The largest difference we observed in human stromal cell preparations was that adipose tissue-derived cells did not exhibit the same mitogenic response to PSC-EVs. There is some literature about the 'specificity' of EV uptake by certain cells (reviewed in *Mulcahy et al., 2014*). For example, pancreatic adenocarcinoma-derived EVs are efficiently internalized by peritoneal cells as opposed to other cell types (*Zech et al., 2012*). In contrast, other studies have shown that fluorescently labeled EVs are internalized by essentially every tested cell type (*Svensson et al., 2013*). This cell type specificity may well have relevance in bone repair, in which multiple cell sources are present which could directly participate in ossification. Whether any recipient cell specificity has to do with relative IGSF8 and PTGFRN cell surface expression or responsiveness to PSC-EV cargo is an intriguing and unanswered question. Of note, PTGFRN expression has been observed in both 'MSCs' (*Billing et al., 2016*) and osteoblasts (*Hanagata and Li, 2011*). Although not specifically addressed in the present study, endothelial cells within a healing bone defect are likely beneficial recipients of EV effects. In prior studies, mesenchymal progenitor cell-derived EVs have been shown to induce a vasculogenic response to aid in wound healing (*Furuta et al., 2016*). Likewise, the human PSCs have been shown to induce vasculogenic responses during wound and bone healing (*Askarinam et al., 2013*; *Bodnar et al., 2016*). Therefore, it is likely that in vivo endothelial cells also receive and respond to perivascular EV signals.

Our in vitro studies suggest that PSCs elaborate large quantities of EVs under standard culture conditions. It is intriguing to hypothesize whether AT-derived PSCs secrete EVs to this degree in vivo under either physiologic or pathologic contexts. Pericytes are intimately associated with their underlying endothelium, we have recently observed that enzymatic dissociation of this relationship significantly changes the pericyte secretome (*Vezzani et al., 2018*). In our past study, Lipogems technology was used to keep the pericyte:endothelial cell contact intact while enzymatic dissociation and FACS purification removed the endothelial cells from culture. Key differences in angiogenic proteins were observed (*Vezzani et al., 2018*), and it is likely that the vesicular secretome was likewise changed. Although likewise speculative, PSC-derived EVs may play roles in pathologic conditions. Blood vessel wall mineralization and ossification may be examples of the unintended consequences of perivascular EVs. The concept of EV-mediated vascular calcification has been explored (*Kapustin et al., 2015*; *Kapustin and Shanahan, 2016*), although the cell of origin has been presumed to be a smooth muscle cell rather than mesenchymal progenitor cell (*Kapustin et al., 2015*). These in vivo correlates represent interesting future avenues of investigation.

Finally, several features of EVs make them an appealing product for clinical translation as compared to their parent MSC. For example, EVs represent a potential 'off the shelf' product which can be stored long term without compromising their activity (*Willis et al., 2017*). According to leading opinions within ISEV, all EV-based therapies will be considered in accordance with the guidelines for biopharmaceutical drug products (*Lener et al., 2015*; *Witwer et al., 2013*), and may move more quickly through regulatory review. Improvement in EV isolation are being developed, such as the continuous centrifugation method, with advantages of convenient and rapid use, low cost, and easy operation. EVs also have low immunogenicity and toxicity but high permeability and biocompatibility (*Sterzenbach et al., 2017*). Thus, whether derived from PSC or other mesenchymal progenitor cell preparations, EV-based therapies represent an attractive future drug for the promotion of tissue repair.

## Summary

In sum, perivascular EVs induce proliferation, migration and osteogenic differentiation of osteoprogenitor cells, and the confluence of these effects positively regulates bone defect repair. Perivascular EVs require surface-associated tetraspanins for bioactivity, and recipient skeletal cells require their binding partners to respond to a perivascular EV stimulus. These data solidify the pleiotropic paracrine effects of perivascular stem cells on bone repair, and suggest the importance of perivascular EVs in both endogenous bone repair as well as in skeletal tissue engineering. Future studies must

consider the issues of process optimization, including the optimum method for PSC-EV isolation, storage, and sustained delivery.

## Materials and methods

### Adipose stem/stromal cell (ASC) and perivascular stem/stromal (PSC) cell isolation

PSCs were isolated from human subcutaneous adipose tissue via fluorescence activated cell sorting (FACS) based on prior protocols (*Hardy et al., 2017*; *Meyers et al., 2018b*). Human lipoaspirate was obtained from healthy adult donors (four different donors) under IRB approval at JHU with a waiver of informed consent, and was stored for less than 48 hr at 4°C before processing. The SVF (stromal vascular fraction) of human lipoaspirate was obtained by type II collagenase digestion. Briefly, lipoaspirate was diluted with an equal volume of phosphate-buffered saline (PBS) and digested with Dulbecco's modified Eagle's medium (DMEM) containing 0.5% bovine serum albumin (Sigma-Aldrich, St. Louis, MO) and 1 mg/ml type II collagenase (Worthington Biochemical, Freehold, NJ) for 1 hr under agitation at 37°C. Adipocytes were separated and removed by centrifugation. The cell pellet was resuspended in red blood cell lysis buffer (155 mM $NH_4Cl$, 10 mM $KHCO_3$, and 0.1 mM EDTA) and incubated at room temperature for 5 min. After centrifugation, cells were resuspended in PBS and filtered at 70 µm. The resuspended SVF cells were cultured in T75 flask and expanded as ASCs. For PSC isolation, the resulting SVF was further processed for FACS sorting, using a mixture of the following directly conjugated antibodies: anti-CD34-allophycocyanin (1:100, RRID:AB_398614; BD Pharmingen, San Diego, CA), anti-CD45-allophycocyanin-cyanin 7 (1:30, RRID: AB_396891; BD Pharmingen), anti-CD146-fluorescein isothiocyanate (1:100, RRID:AB_324069; Bio-Rad, Hercules, CA), and anti-CD31-allophycocyanin-cyanin 7 (1:100, RRID:AB_10643590; Bio Legend, San Diego, CA). A summary of antibodies used is presented in *Supplementary file 10*. All incubations were performed at 4°C for 20 min. The solution was then passed through a 40 µm cell filter and then run on a FACS Diva 8.0.1 cell sorter (BD Biosciences). FlowJo software (version 7.6, RRID:SCR_008520) was used for the analysis of flow cytometry data. In this manner, microvessel pericytes (CD146+CD34-CD45-CD31-) and adventitial cells (CD34+CD146-CD45-CD31-) were isolated and combined to constitute the PSC population. Each patient sample of cells was cultured independently. For in vitro experiments, cells were cultured at 37°C in a humidified atmosphere containing 95% air and 5% $CO_2$. ASCs and PSCs were expanded in growth medium (GM) consisting of DMEM, 15% fetal bovine serum (FBS) (Gibco, Grand Island, NY), 1% penicillin/streptomycin (Gibco). Medium was changed every 3 day unless otherwise noted.

### Bone marrow mesenchymal stem cell (BMSC) isolation and expansion

Bone marrow mesenchymal cells (BMSCs) of de-identified arthroplasty specimens of the human femur and tibia were flushed with PBS. All samples were obtained under IRB approval at JHU with a waiver of informed consent. Marrow cells were passed through a 70 µm cell strainer (BD Bioscience) to obtain a single-cell suspension of all nucleated cells. BMSCs were expanded in growth medium as above. Non-adherent cells were removed by washing the cultures with PBS twice and replacing the medium after 5 days. Multilineage differentiation capacity of human BMSCs was confirmed using previously reported methods, including osteogenic differentiation and Alizarin red staining (*Xu et al., 2016*), adipogenic differentiation and Oil red O staining (*Barlian et al., 2018*), and chondrogenic differentiation in high density micromass and Alcian blue staining (*Huang et al., 2014*). Briefly, differentiation protocols were as follows: For osteogenic or adipogenic differentiation, after reaching 90% confluency in monolayer, growth medium was replaced with either osteogenic induction medium (100 nM dexamethasone, 50 µM ascorbic acid, and 10 mM β-glycerophosphate (Sigma-Aldrich)) or adipogenic induction medium (MesenCult Adipogenic Differentiation Medium (Human; Stemcell Technologies, Vancouver, Canada)). In high-density micromass, chondrogenic induction medium was used (MesenCult-ACF Chondrogenic Differentiation Medium (Stemcell Technologies)). Human BMSCs were cultured and used for experiments at passage three unless otherwise noted.

## Mouse calvarial cell and adipose-derived stromal/stem cell isolation

Neonatal mouse calvarial cells (NMCCs) were collected from C57BL/6J mice at postnatal day 1. Dissected frontal and parietal bones were subjected to six sequential enzymatic digestions with collagenase type I (Worthington Biochemical; 1 mg/mL) and collagenase type II (Worthington Biochemical; 1 mg/mL). The dissociated cells (from sequential digestions 3–5) were then filtered through a 40 μm strainer and cultured in α-MEM supplemented with 15% FBS and 1% penicillin/streptomycin.

For isolation of murine adipose-derived stromal/stem cells, subcutaneous fat pads were excised separately from C57BL/6J mice, finely minced, and digested using collagenase type II (Worthington Biochemical; 0.75 mg/mL) for 30 min at 37˚C. The cell suspension was filtered through a 70 μm strainer and centrifuged at 1000 rpm for 5 min. The cells were plated in T75 flask and cultured in α-MEM supplemented with 15% FBS and 1% penicillin/streptomycin.

## PSC: BMSC co-culture assays and analysis

Co-culture experiments were performed using 24-well 0.4 um transwell inserts (Millipore, Darmstadt, Germany) with human PSCs (passage 8) placed in the upper insert and human BMSCs (passage 3) in the lower well. Proliferation was measured after 72 hr co-culture in GM using the CellTiter96 AQueous One Solution Cell Proliferation Assay kit (MTS, G358A; Promega, Madison, WI), where $8 \times 10^3$ BMSCs were cultured with or without $3 \times 10^5$ PSCs. Briefly, 20 μl of MTS solution was added to each well. After incubation for 1 hr at 37˚C, the absorbance was measured at 490 nm using Epoch microspectrophotometer (Bio-Tek, Winooski, VT). Cell migration was measured with Ibidi inserts (Ibidi, Planegg/Martinsried, Germany), where $1.4 \times 10^4$ BMSCs were seeded and grown to confluency followed by co-culture with or without $3 \times 10^5$ PSCs in reduced FBS conditions (GM with 1% FBS). Inserts were removed and cell migration into the empty area was monitored by brightfield microscopy at 0, 8, or 24 hr. The equilibrium width of the gap was calculated using the ImageJ software (Version 1.49 v, RRID:SCR_003070; NIH, Bethesda, MD). Here, gap closure = (scratch width at hour 0 − scratch width at hour 8 or 24)/scratch width at hour 0 × 100%. Osteogenic differentiation in co-culture was performed using osteogenic differentiation medium (OM) consisting of GM with 50 μM ascorbic acid, 10 mM β-glycerophosphate, and 100 nM dexamethasone (Sigma-Aldrich). $4 \times 10^4$ BMSCs were cultured with or without of $5 \times 10^5$ PSCs, and alkaline phosphatase (ALP) staining was performed at 72 hr using an ALP staining kit (Sigma-Aldrich). Relative staining intensity was quantified using ImageJ software and normalized to the control group. In select studies, PSCs were labeled with PKH26 Red Fluorescent Cell Linker Kit (Sigma-Aldrich). After 48 hr co-culture of PKH26-labeled PSCs ($5 \times 10^5$) with unlabeled BMSCs ($2 \times 10^4$), images were acquired with an Olympus IX71 inverted microscope (Olympus, Cypress, CA). Unless otherwise stated, each experimental study was done in three technical replicates using two different PSC preparations.

## Extracellular vesicles isolation and analysis

EVs were derived from passage 3–9 cells using ultracentrifugation based on previously validated protocols (*Sung et al., 2015*; *Théry et al., 2006*). Each preparation of EVs was isolated from independent cells and were not mixed. Briefly, ASCs or PSCs were expanded in growth medium. Upon reaching subconfluency and after triplicate washes in PBS, cells were cultured in DMEM only for 48 hr. EVs were collected by serial centrifugation at $300 \times g$ for 10 min, $2\,000 \times g$ for 30 min, $10\,000 \times g$ for 30 min, and $120\,000 \times g$ for 4 hr at 4˚C. The supernatant was discarded and the pellets were resuspended in 1X phosphate-buffered saline (PBS). The protein concentration of EVs were quantified using the Pierce BCA Protein Assay Kit (Thermo Scientific, Waltham, MA), according to the manufacturer's instruction in similarity to prior reports (*Li et al., 2016*). For transmission electron microscopy (TEM), 10 μl of sample was adsorbed to glow-discharged 400 mesh ultra-thin carbon coated grids (EMS CF400-CU-UL) for two min, followed by three quick rinses of TBS and staining with 1% UAT (uranyl acetate with 0.05 Tylose). Grids were immediately observed with a Philips CM120 at 80 kV and images captured with an AMT XR80 high-resolution (16-bit) 8 Mpixel camera. The size distribution of EVs was examined by analysis of serial TEM images (n = 57 images analyzed) or using nanoparticle tracking analysis (NTA) with NanoSight NS500 (Malvern, Worcestershire, UK) with a 405 nm laser. For Western blotting, cells were lysed in RIPA buffer (Thermo Scientific) with protease inhibitor cocktail (Cell Signaling Technology, Danvers, MA). Proteins were separated by SDS–polyacrylamide gel electrophoresis and transferred onto a nitrocellulose membrane. Protein

extraction was blocked with 5% bovine serum albumin and incubated with primary antibodies at 4°C overnight. Finally, membranes were incubated with a horseradish-peroxidase (HRP)-conjugated secondary antibody and detected with ChemiDoc XRS+ System (Bio-rad).

## EV in vitro assays

EVs (1, 2.5, or 5 µg/mL protein content) were added to culture medium during proliferation, migration and osteogenic differentiation as described above. The ratio of particles to protein was $1.18*10^9$ particles/µg. In select experiments, PSC-EVs were labeled using PKH26 lipophilic dye or a pH sensitive dye (pHrodo Red Maleimide, Thermo Scientific) prior to use. Upon completion of the reaction with the PSC-EVs, an excess of glutathione was added to consume the excess thiol-reactive reagent. PSC-EVs and dye were separated by a gel filtration column (Sephadex G-25 column; GE Healthcare, Marlborough, MA). Proliferation assays were performed in 96-well plates ($2 \times 10^3$ BMSCs/well) and assayed at 48, 72, and 96 hr. Migration was performed as above with or without EVs supplementation (1, 2.5, or 5 µg/mL), with endpoints at 8 and 24 hr. Osteogenic differentiation was performed with OM with or without EVs (1, 2.5, or 5 µg/mL), with medium and EVs replenished every 3 days. The degree of mineralization was assessed by Alizarin Red S staining after 7 days (Sigma-Aldrich), followed by incubation with 0.1 N sodium hydroxide and photometric quantification using Epoch microspectrophotometer (548 nm absorbance). Total RNA was extracted from the cultured cells using TRIzol Reagent (Invitrogen, Carlsbad, CA) according to the manufacturer's instructions. 0.8 µg of total RNA was used for reverse transcription with iScript cDNA synthesis kit (Bio-Rad) following manufacturer's instructions. Real-time PCR was performed using SYBR Green PCR Master Mix (Thermo Scientific) according to the manufacturer's protocol. Relative gene expression was calculated using a $2^{-\Delta\Delta Ct}$ method by normalization with GAPDH. Primer sequences are presented in *Supplementary file 11*. In select experiments, PSC-EVs were incubated with trypsin (1 mg/ml, Sigma) for 1 hr at 37°C and re-isolated by ultracentrifugation prior to application. In select experiments, PSC-EVs were pre-incubated with anti-CD9 (RRID:AB_302894) or anti-CD81 (RRID:AB_2811127) neutralizing antibodies, or their appropriate IgG isotype control prior to application. For neutralizing antibody experiments, 2.5 µg PSC-EVs was incubated with 10 µl neutralizing antibody or isotype control (Anti-CD9: 1 mg/mL; Anti-CD81: 0.88 mg/mL) for 4 hr at 4°C prior to use.

## shRNA knockdown

In select experiments, shRNA-mediated knockdown of the CD9/CD81 receptors PTGFRN and IGSF8 was performed among primary human BMSCs prior to PSC-EVs application. ShRNA was transfected using TransIT-LT1 Transfection Reagent (Mirus Bio, Madison, WI) as described by the manufacturer. The target sequence of *PTGFRN* mRNA was 5'- GCCTTTGATGTGTCCTGGTTT-3' and that of *IGSF8* mRNA was 5'-GCTGCTGCTAATGCTAGGAAT-3'. The medium was changed after 4 hr. Validation by regular PCR and semi-quantification was performed in ImageJ software.

## Transcriptomics

The RNA content of PSC-EVs and parent PSCs was detected by total RNA sequencing. Briefly, total RNA was extract from PSCs by Trizol (Life technologies corporation). PSC-EV-derived RNA was isolated using exoRNeasy Serum Plasma Kits (Qiagen, Hilden, Germany) in accordance with the manufacturer's instructions. The RNA samples were sent to the JHMI Deep Sequencing and Microarray core (JHU) and quantified by deep sequencing with the Illumina NextSeq 500 platform (Illumina, San Diego, CA). Data analyses were performed using software packages including CLC Genomics Server and Workbench (RRID:SCR_017396 and RRID:SCR_011853), Partek Genomics Suite (RRID:SCR_011860), Spotfire DecisopnSite with Functional Genomics (RRID:SCR_008858), and QIAGEN Ingenuity Pathway Analysis (RRID:SCR_008653).

The transcriptome of BMSCs treated with PSC-EVs or PBS control for 48 hr was examined by expression microarray. Briefly, total RNA was extracted from BMSCs by Trizol (Life technologies corporation, Gaitherburg, MD, USA). The RNA samples were sent to the JHMI Deep Sequencing and Microarray core (JHU, Baltimore, MD) and quantified by microarray analyses on the Affymetrix Clariom_D Array (Affymetrix, Santa Clara, CA). Data analyses were performed using software packages including CLC Genomics Server and Workbench (RRID:SCR_017396 and RRID:SCR_011853), Partek

Genomics Suite (RRID:SCR_011860), Spotfire DecisionSite with Functional Genomics (RRID:SCR_008858), and QIAGEN Ingenuity Pathway Analysis (RRID:SCR_008653).

## Animal care and surgical procedures

All animal experiments were performed according to the approved protocol of the Animal Care and Use Committee (ACUC) at Johns Hopkins University (Approval No. MO16M226). 10-week-old, male C57BL/6J mice were purchased from the Jackson Laboratory (strain #000664, RRID:IMSR_JAX:000664, Bar Harbor, ME). To assess progenitor cell migration, 10-week-old, male Pdgfrα-CreER; eGFP transgenic reporter mice were a kind gift from the Dwight Bergles laboratory (*Kang et al., 2010*). Validation of the high specificity of Pdgfrα reporter activity has been previously confirmed (*Kang et al., 2010*). In Pdgfrα-CreER;eGFP transgenic reporter mice, tamoxifen (TM) administration was performed by i.p. injection as per published protocols 14 days prior to defect creation (*Kang et al., 2010*). TM free base (MP Biomedicals, Solon, OH) in 98% sunflower seed oil and 2% ethanol was administered at a concentration of 0.07 mg/kg daily for 5 days by i.p. injection as per previously validated protocols (*Kang et al., 2010*). For calvarial defect creation, anesthesia was performed with 2–3% isoflurane in 100% oxygen at a flow rate of 1 L/min and animals were operated upon on a warm, small animal surgery station. Post-operative monitoring was performed in accordance with institutional policy. Analgesia was administered using buprenorphine (0.1 ml/25 g body weight) via intraperitoneal injection after surgery, and a repeat injection was performed after 48 hr. A 4 mm skin incision was made over the right frontal bone. Next, a 1.8 mm diameter full thickness circular defect was created in the non-suture associated frontal bone using an Ideal Micro-Drill and a burr (Xemax Surgical, Napa Valley, CA). Meticulous care was taken not to injure the underlying dura mater. Finally, the skin was sutured and the animal was monitored per established postoperative protocols. PSC-EVs (1 or 2.5 μg total dose) or vehicle control were percutaneously injected into the tissue directly overlying the defect every 3 days (27G × 5/8' Insulin Syringes, BD, Franklin Lakes, NJ). Mice were euthanized after 4 weeks for postmortem analysis.

## Postmortem analyses

Samples were fixed in 4% PFA (paraformaldehyde) for 24 hr and imaged using a high-resolution microcomputed tomography (microCT) imaging system (SkyScan 1294; Bruker MicroCT N.V, Kontich, Belgium). Scans were obtained at an image resolution of 10 μm, with the following settings: 1 mm of aluminum filter, X-ray voltage of 65 kVP, anode current of 153 uA, exposure time of 65 ms, frame averaging of 4, and rotation step of 0.3 degrees. Three-dimensional images were then reconstructed from the 2D X-ray projections by implementing the Feldkamp algorithm using a commercial software package NRecon software (2.0.4.0 SkyScan, Bruker). For the 3D morphometric analyses of images, CTVox and CTAn software were used (1.13 SkyScan, Bruker). For calvarial defect analysis, a cylindrical volume of interest centered around each defect site was defined as the 1.8 mm in diameter and 1 mm in height with a threshold value of 80. The amount of bone formation was analyzed and quantified in three different ways. Firstly, bone volume (BV) was calculated from binary x-ray images. Second, bone fractional area (BFA) was calculated by using CTVox to create a 3D rendering of calvarial defect and measuring pixels of bone in defect divided by total defect area using Adobe Photoshop (RRID:SCR_014199; Adobe, San Jose, CA). Lastly, a bone healing score from 0 to 4 was assigned by three blinded observers according to previous published grading scales for calvarial defect healing (*Spicer et al., 2012*). Briefly, the grading system was as follows: 0–no bone formation, 1–few bony spicules dispersed through defect, 2–bony bridging only at defect borders, 3–bony bridging over partial length of defect, and 4–bony bridging entire span of defect at longest point.

After radiographic imaging, samples were transferred to 14% EDTA for decalcification for 14–21 days. Samples were then embedded in optimal cutting temperature compound (OCT) and sectioned in a coronal plane at 10 μm thickness. H&E staining was performed on serial sections. For immunofluorescent staining, additional sections were incubated with the following primary antibodies: anti-Ki67 (1:200, RRID:AB_302459), and anti-Osteocalcin (1:100, RRID:AB_10675660). Sections were washed with phosphate buffered saline (PBS) three times, 10 min each. All sections were blocked with 5% goat serum in PBS for 1 hr at 25°C; antigen retrieval was by trypsin enzymatic antigen retrieval solution for 10 mins at 37°C (ab970; Abcam, Cambridge, MA). Primary antibodies were added to each section at their respective dilutions and incubated at 37°C for 1 hr and then overnight

at 4°C. Next, a Dylight 594 goat anti-rabbit IgG (H+L) polyclonal (1:200, RRID:AB_2336413) was used as the secondary antibody. Sections were counterstained with DAPI mounting medium (H-1500, Vector laboratories, Burlingame, CA). For studies in Pdgfrα-CreER;eGFP mice, histologic preparations were examined at 7 days post-injury, and total reporter activity within 3–4 random high-powered images per defect site were quantified using ImageJ software. All histological sections were examined under a Zeiss 700 confocal microscope (Zeiss, Thornwood, NY).

## Statistical analysis

Quantitative data are expressed at mean ± SD. Statistical analyses were performed using the SPSS16.0 software (RRID:SCR_002865). All data were normally distributed. Student's t test was used for two-group comparisons, and one-way ANOVA test was used for comparisons of three or more groups, followed by Tukey's post hoc test. Differences were considered significant when $*p<0.05$, $**p<0.01$, and $***p<0.001$.

## Data availability

Expression data that support the findings of this study have been deposited in Gene Expression Omnibus (GEO) with the accession codes GSE118961 and GSE130086.

## Acknowledgements

AWJ was supported by the NIH/NIAMS (R01 AR070773, K08 AR068316), NIH/NIDCR (R21 DE027922), Department of Defense (W81XWH-18-1-0121, W81XWH-18-1-0336, W81XWH-18–10613), American Cancer Society (Research Scholar Grant, RSG-18-027-01-CSM), the Orthopaedic Research and Education Foundation with funding provided by the Musculoskeletal Transplant Foundation, the Maryland Stem Cell Research Foundation, and the Musculoskeletal Transplant Foundation. The content is solely the responsibility of the authors and does not necessarily represent the official views of the National Institute of Health or Department of Defense. We thank the JHU microscopy facility and the Deep Sequencing and Microarray Core for their technical and analytical assistance.

## Additional information

### Competing interests

Bruno Peault: is the inventor of perivascular stem cell-related patents held by the UC Regents (Patent No. 20160271186). Aaron Watkins James: is a scientific advisory board member for Novadip, LLC. The other authors declare that no competing interests exist.

### Funding

| Funder | Grant reference number | Author |
|---|---|---|
| National Institute of Arthritis and Musculoskeletal and Skin Diseases | R01 AR070773 | Aaron Watkins James |
| National Institute of Dental and Craniofacial Research | R21 DE027922 | Aaron Watkins James |
| Department of Defense | W81XWH-18-1-0121 | Aaron Watkins James |
| American Cancer Society | Research Scholar Grant RSG-18-027-01-CSM | Aaron Watkins James |
| Orthopaedic Research and Education Foundation | | Aaron Watkins James |
| Maryland Stem Cell Research Fund | | Aaron Watkins James |
| Musculoskeletal Transplant Foundation | | Aaron Watkins James |

| National Institute of Arthritis and Musculoskeletal and Skin Diseases | K08 AR068316 | Aaron Watkins James |
|---|---|---|
| Department of Defense | W81XWH-18-1-0336 | Aaron Watkins James |
| Department of Defense | W81XWH-18-10613 | Aaron Watkins James |

The funders had no role in study design, data collection and interpretation, or the decision to submit the work for publication.

## Author contributions

Jiajia Xu, Data curation, Formal analysis, Writing—original draft; Yiyun Wang, Ching-Yun Hsu, Yongxing Gao, Carolyn Ann Meyers, Leslie Chang, Leititia Zhang, Catherine Ding, Data curation, Formal analysis; Kristen Broderick, Bruno Peault, Kenneth Witwer, Provision of study material; Aaron Watkins James, Funding acquisition, Writing—original draft, Project administration, Writing—review and editing, Conception and design

## Author ORCIDs

Jiajia Xu (iD) https://orcid.org/0000-0002-6084-2029
Aaron Watkins James (iD) https://orcid.org/0000-0002-2002-622X

## Ethics

Human subjects: Human lipoaspirate was obtained under IRB approval at JHU with a waiver of informed consent (Approval No. IRB00119905 and IRB00137530).
Animal experimentation: All animal experiments were performed according to the approved protocol of the Animal Care and Use Committee (ACUC) at Johns Hopkins University (Approval No. MO16M226).

## Decision letter and Author response

Decision letter https://doi.org/10.7554/eLife.48191.035
Author response https://doi.org/10.7554/eLife.48191.036

# Additional files

## Supplementary files

• Supplementary file 1. Key Resources Table.
DOI: https://doi.org/10.7554/eLife.48191.017

• Supplementary file 2. Frequency of human PSC in lipoaspirate used.
DOI: https://doi.org/10.7554/eLife.48191.018

• Supplementary file 3. Yields of PSC-EV. Yield produced by each PSC-EV isolation are summarized. Protein amounts of harvested PSC-EV were determined by the BCA method. For patient samples 1 and 2, the same cell population was used to isolate EVs at two different passages as indicated.
DOI: https://doi.org/10.7554/eLife.48191.019

• Supplementary file 4. Highest 100 transcripts in human PSC-EVs.
DOI: https://doi.org/10.7554/eLife.48191.020

• Supplementary file 5. Relative gene expression within human PSC-EVs or parent PSC among transcription factors enriched in porcine ASC-EVs.
DOI: https://doi.org/10.7554/eLife.48191.021

• Supplementary file 6. Most upregulated pathways among PSC-EV-treated BMSC by Ingenuity Pathway Analysis.
DOI: https://doi.org/10.7554/eLife.48191.022

• Supplementary file 7. Most downregulated pathways among PSC-EV-treated BMSC by Ingenuity Pathway Analysis.
DOI: https://doi.org/10.7554/eLife.48191.023

• Supplementary file 8. CD markers enriched in PSC-EVs.
DOI: https://doi.org/10.7554/eLife.48191.024

• Supplementary file 9. Animal allocation and treatment groups.
DOI: https://doi.org/10.7554/eLife.48191.025

• Supplementary file 10. Antibodies used.
DOI: https://doi.org/10.7554/eLife.48191.026

• Supplementary file 11. Quantitative PCR primers used.
DOI: https://doi.org/10.7554/eLife.48191.027

• Supplementary file 12. Basic features of human PSCs, ASCs, and BMSCs.
DOI: https://doi.org/10.7554/eLife.48191.028

• Transparent reporting form
DOI: https://doi.org/10.7554/eLife.48191.029

## Data availability

Sequencing data have been deposited in GEO under accession codes GSE118961 and GSE130086.

The following datasets were generated:

| Author(s) | Year | Dataset title | Dataset URL | Database and Identifier |
|---|---|---|---|---|
| Aaron James, Jiajia Xu | 2018 | Expression data from human bone marrow mesenchymal stem cells – treated with perivascular stem cell-derived extracellular vesicles control (in osteogenic medium) | https://www.ncbi.nlm.nih.gov/geo/query/acc.cgi?acc=GSE118961 | NCBI Gene Expression Omnibus, GSE118961 |
| Aaron James, Jiajia Xu | 2019 | Expression data from human perivascular stem cell-derived extracellular vesicles (PSC-EVs) and PSCs | https://www.ncbi.nlm.nih.gov/geo/query/acc.cgi?acc=GSE130086 | NCBI Gene Expression Omnibus, GSE130086 |

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
