## [Decision Letter]

Thank you for submitting your article "Human perivascular stem cell-derived extracellular vesicle mediate bone repair requiring surface-associated tetraspanins" for consideration by *eLife*. Your article has been reviewed by Utpal Banerjee as the Senior Editor, a Reviewing Editor, and three reviewers. The following individuals involved in review of your submission have agreed to reveal their identity: Hadil al-Jallad (Reviewer #1).

Overall there was general enthusiasm for the manuscript and the revision should be feasible within a short period of time.

Summary:

Overall there was strong support for the hypothetical construct, the experimental results and the overall conclusions that EVs from vascular pericytes may mediate bone repair.

Essential revisions:

Title: There was some concern that the title may be a bit over-reaching in respect to the tetraspanins as the sole mediator of the repair process; hence a change in the title is indicated.

In the revised manuscript, it would be helpful to address these concerns:

1) The role of other CD markers in the pro-osteogenic effect.

2) Clearer contrast of the ASC vs. PSC-EVs pro-osteogenic actions.

3) More fully justify the human PSC-EV effect on calvariae from B6 mice in respect to species specificity and the actions of human CD9 and CD81 on murine receptors.

Comments from the three reviewers are noted below to assist in your revision:

*Reviewer #1:*

This is a well written and well referenced manuscript. It adds a valuable evidence to the pro-bone regenerative properties of some cell-specific EVs. Nevertheless, the title currently used by the authors implies that EVs with certain tetraspanins mediate bone repair.

The experimental design of the bone repair shows that PSC-EVs in general have induced bone repair, which is not specifically associated to the CD markers (tetraspanins under investigation in this study).

My comments are summarized below:

With regard to the processing and selection of the clinical samples: It was not clear to the reviewer of how many replicates/clinical sample? Were the PCSs mixed together after cell sorting as one universal sample? Or were they treated as independent samples? Accordingly, what is the source of the elaborated EVs? Were the EVs isolated from independent PSCs representing each clinical sample grown in culture, or was the source a collective mix of PSCs?

Figure 2:

Alkaline phosphatase activity or qPCR for ALP gene expression are better read out assays to manifest the PSC-EVs induced BMSC ECM osteogenesis.

Figure 3:

Figure 3H through 3K: two extra controls should be included, BMSC without IgG and BMSC incubated with PSC-EV without IgG within the same experiential setting. Figure 3K, why there is a difference between AZ reading among the IgG group and CD81 NAB group? One would expect the reading to be the same in both controls.

Figure 4E:

Knocking down the receptors of CD9 and CD81 has no effect on osteogenesis, mineralization in both the experimental conditions and controls were comparable. This possibly indicates that other EV-CD markers are involved in promoting osteogenesis and not necessarily dependent or mediated by CD9 or CD81 only. Based on the transcriptome study, what other CD markers could specifically contribute to the pro-osteogenic effect?

Figure 5:

This experiment and the proof that PSC-EV promote bone repair in a mouse model of critical size calvarial defect does not indicate specifically that only CD9, and CD81 positive EVs have contributes to the bone repair. EVs are heterogeneous in nature, we do not know which EVs have contributed to the bone repair. This is difficult to claim at this point since the technical tools available to the EV community cannot isolate CD-specific EVs and link them to bone repair.

*Reviewer #2:*

This is a well written, well referenced and well designed study. The work confirms and extends the existing literature from the authors' and other's laboratories. The findings demonstrate that both pericytes or their exosomes/microvesicles can be used to enhance osteogenesis and wound healing in vitro and in vivo. The findings have broad interest to the adipose, bone, and vascular biology fields.

Essential revisions:

1) The authors discuss the concept of exosomes as an off the shelf substitute for actual pericytes in a clinical translational setting. The authors are encouraged to expand on this concept in the Discussion. Please highlight how exosome products could be easier to move through regulatory review, cheaper to manufacture, and less likely to elicit an immune response.

2) The work focuses almost exclusively on the FACS purified homogeneous pericyte and adventitial cell populations from adipose tissue. The process of isolating such cells adds considerable expense to the manufacturing process. Did the authors have any data on heterogenous, unsorted adipose stromal/stem cells alone? The inclusion of such data, even in a limited manner, in comparison to the isolated pericytes and adventitial cells using at least the in vitro assays, would provide a basis for the continued use and focus on the pericytic population as a source of commercial clinical translational products and manufacture.

*Reviewer #3:*

Authors are here proposing an extremely interesting approach where adipose derived cells release EV that in turn influence marrow stromal cells performance in vitro in a dose dependent manner while same impact is difficult the be achieved using adipose stromal cells as target. They also propose that EV effects is dependent on surface-associated tetraspanins CD9 or CD81 via their binding partners IGSF8 and PTGFRN mostly by in vitro experiments. There are few very major aspects I would suggest to take into account:

1) A summarizing table with data to outline the features (isolation/culture techniques, basic morphology, FACS phenotype and cell culture doublings) of human PSC, BMSC and ASC is needed to clarify better all these different mesenchymal cell populations used as target and effectors.

2) Authors should justify/discuss why the impact of PSC-EVs on BMSCs is greater than with ASC. Any difference in the transcriptome level for targets BMCS versus ASC?

3) Why do the authors investigate whether human PSC-EV treatment may improve calvarial defect in BL6 mice. This is bit odd and does not account species specificity not only in the genomic signals exchange but also in the CD9/CD81 interaction with their murine receptors. To validate this model they should take marrow and or adipose cells from mice and treat them with human PSC-EV in vitro as done for human samples using the same read-out. We shall expect similar outcome.

4) In subsection “PSC-EVs promote BMSC proliferation, migration, and osteogenic differentiation”: "PSC-EVs were next labeled with a pH dependent dye" Please specify this better.

5) In Figure 1C, the ALP staining is not very convincing. Are there any better pictures? Maybe Alizarin Red may help, also to keep consistency.

6) Unfortunately, Figure 5I and 5J are again not very convincing and should be better explained since we are unable to detect significant differences in the groups.

7) Were EV also purified and diluted (not in co-culture) into media culture alone to generate the desired effect just based on EV?

---

## [Author Response]

Essential revisions:Title: There was some concern that the title may be a bit over-reaching in respect to the tetraspanins as the sole mediator of the repair process; hence a change in the title is indicated.

As suggested, the title has been changed to “Human perivascular stem cell-derived extracellular vesicles mediate bone repair.”

In the revised manuscript, it would be helpful to address these concerns:1) The role of other CD markers in the pro-osteogenic effect.2) Clearer contrast of the ASC vs. PSC-EVs pro-osteogenic actions.3) More fully justify the human PSC-EV effect on calvariae from B6 mice in respect to species specificity and the actions of human CD9 and CD81 on murine receptors.

These three issues have been fully addressed. We have (1) revisited our transcriptome data to identify other potential cell surface markers of interest in EV-BMSC interaction and the resulting pro-osteogenic effects, (2) performed additional experiments that more clearly compare ASC-EV vs. PSC-EV, and (3) performed additional experiments confirming the effects of PSC-EV on murine cells.

Comments from the three reviewers are noted below to assist in your revision:Reviewer #1:My comments are summarized below:With regard to the processing and selection of the clinical samples: It was not clear to the reviewer of how many replicates/clinical sample? Were the PCSs mixed together after cell sorting as one universal sample? Or were they treated as independent samples? Accordingly, what is the source of the elaborated EVs? Were the EVs isolated from independent PSCs representing each clinical sample grown in culture, or was the source a collective mix of PSCs?

The number of replicates and number of clinical samples for each study has been more clearly reported. All EVs were independently isolated from ASC or PSC samples, and were not mixed (Materials and methods section).

Figure 2:Alkaline phosphatase activity or qPCR for ALP gene expression are better read out assays to manifest the PSC-EVs induced BMSC ECM osteogenesis.

Agree. This has been changed as suggested (subsection “PSC-EVs promote BMSC proliferation, migration, and osteogenic differentiation” and Figure 2E).

Figure 3:Figure 3H through 3K: two extra controls should be included, BMSC without IgG and BMSC incubated with PSC-EV without IgG within the same experiential setting. Figure 3K, why there is a difference between AZ reading among the IgG group and CD81 NAB group. One would expect the reading to be the same in both controls.

Agree. We examined these two controls in both Figure 2F and Figure 3—figure supplement 1C. IgG did not significantly affect BMSC osteogenesis. Indeed, anti-CD81 antibody alone did inhibit BMSC osteogenic differentiation to some degree (Figure 3K). Although interesting, the potential direct effects of anti-CD81 without EVs was not the main subject of investigation and was not further pursued.

Figure 4E:Knocking down the receptors of CD9 and CD81 has no effect on osteogenesis, mineralization in both the experimental conditions and controls were comparable. This possibly indicates that other EV-CD markers are involved in promoting osteogenesis and not necessarily dependent or mediated by CD9 or CD81 only. Based on the transcriptome study, what other CD markers could specifically contribute to the pro-osteogenic effect?

Thank you for the good point. Our observations found that EV induced osteogenic differentiation required the CD9/81 receptor PTGFRN but not IGSF8 on the recipient cell. Toward your good point, we returned to our transcriptome to examine potential other cell surface proteins that may be important in EV internalization (such as *CD46* or *CD63*) or otherwise mediate PSC-EV pro-osteogenic effects (such as *CD44, CD82*, and *CD99*). These interesting findings are further discussed in the revision (subsection “PSC-EVs induce proliferation, migration, and osteogenic differentiation to induce calvarial defect healing”) and shown in new Supplementary file 8.

Figure 5:This experiment and the proof that PSC-EV promote bone repair in a mouse model of critical size calvarial defect does not indicate specifically that only CD9, and CD81 positive EVs have contributes to the bone repair. EVs are heterogeneous in nature, we do not know which EVs have contributed to the bone repair. This is difficult to claim at this point since the technical tools available to the EV community cannot isolate CD-specific EVs and link them to bone repair.

Agree. In accordance with your suggestion, we have modified the title of our revision. In the future, we hope to have the technical tools to further investigate CD-specific EVs.

Reviewer #2:Essential Revisions:1) The authors discuss the concept of exosomes as an off the shelf substitute for actual pericytes in a clinical translational setting. The authors are encouraged to expand on this concept in the Discussion. Please highlight how exosome products could be easier to move through regulatory review, cheaper to manufacture, and less likely to elicit an immune response.

Thank you. A discussion of these concepts has been added as suggested (Materials and methods section).

2) The work focuses almost exclusively on the FACS purified homogeneous pericyte and adventitial cell populations from adipose tissue. The process of isolating such cells adds considerable expense to the manufacturing process. Did the authors have any data on heterogenous, unsorted adipose stromal/stem cells alone? The inclusion of such data, even in a limited manner, in comparison to the isolated pericytes and adventitial cells using at least the in vitro assays, would provide a basis for the continued use and focus on the pericytic population as a source of commercial clinical translational products and manufacture.

According to your good point, we have compared the effects of ASC-EVs and PSC-EVs on cell proliferation, migration, and osteogenic differentiation. Although both ASC and PSC derived EVs have similar positive regulatory effects on BMSC cell behavior, the magnitude of mitogenic and pro-osteogenic changes is significantly greater among PSC-EV. We concur that FACS purification adds considerable cost, but also believe that the higher homogeneity of a FACS identified population has both ramifications for safety and efficacy of a potential stem cell therapy. Please see revised Figure 2—figure supplement 2 (Results section).

Reviewer #3:Authors are here proposing an extremely interesting approach where adipose derived cells release EV that on turn influence marrow stromal cells performance in vitro in a dose dependent manner while same impact is difficult the be achieved using adipose stromal cells as target. They also propose that EV effects is dependent on surface-associated tetraspanins CD9 or CD81 via their binding partners IGSF8 and PTGFRN mostly by in vitro experiments. There are few very major aspects I would suggest to take into account:1) A summarizing table with data to outline the features (isolation/culture techniques, basic morphology, FACS phenotype and cell culture doublings) of human PSC, BMSC and ASC is needed to clarify better all these different mesenchymal cell populations used as target and effectors.

This has been added as suggested (new Supplementary file 12).

2) Authors should justify/discuss why the impact of PSC-EVs on BMSCs is greater than with ASC. Any difference in the transcriptome level for targets BMCS versus ASC?

Thank you for the good point. Recipient cell type specificity was one of several findings that, while interesting to pursue, would have significantly expanded the scope of the present study. Instead we focused on specific tetraspanins and their interactions with recipient BMSCs. Cell type heterogeneity in their response to EVs is an interesting future subject of investigation.

3) Why do the authors investigate whether human PSC-EV treatment may improve calvarial defect in BL6 mice. This is bit odd and does not account species specificity not only in the genomic signals exchange but also in the CD9/CD81 interaction with their murine receptors. To validate this model they should take marrow and or adipose cells from mice and treat them with human PSC-EV in vitro as done for human samples using the same read-out. We shall expect similar outcome.

Thank you for the suggestion. Additional studies were performed using mouse recipient cells with the same readouts, including mouse calvarial cells (to directly align with in vivo mouse studies) and mouse ASC as suggested. Results showed that mouse cells are responsive to human PSC-EV in the same manner as human cells. Please see revised Figure 5—figure supplement 1 (subsection “PSC-EVs induce proliferation, migration, and osteogenic differentiation to induce calvarial defect healing”).

4) In subsection “PSC-EVs promote BMSC proliferation, migration, and osteogenic differentiation”: "PSC-EVs were next labeled with a pH dependent dye". Please specify this better.

Agree. This has been clarified (subsection “PSC-EVs promote BMSC proliferation, migration, and osteogenic differentiation”).

5) In Figure 1C, the ALP staining is not very convincing. Are there any better pictures? Maybe Alizarin Red may help, also to keep consistency.

Thank you for the good point. A high magnification image has been added which better illustrates the difference in ALP activity (Figure 1C).

6) Unfortunately, Figure 5I and 5J are again not very convincing and should be better explained since we are unable to detect significant differences in the groups.

Thank you for the suggestion. These images have been updated and enlarged so as to make more clear our findings (Figure 5I,J).

7) Were EV also purified and diluted (not in co-culture) into media culture alone to generate the desired effect just based on EV?

We apologize for the lack of clarity. EVs were purified and diluted in cell culture media in defined concentrations for all studies in Figures 2, 3, and 4 (not in co-culture).